# Chemical Property Changes and Thermal Analysis during the Carbonizing Process of the Pollen Grains of *Typha*

**DOI:** 10.3390/molecules24010128

**Published:** 2018-12-31

**Authors:** Mingliang Gao, Beihua Bao, Yudan Cao, Mingqiu Shan, Fangfang Cheng, Miao Jiang, Peidong Chen, Li Zhang

**Affiliations:** Jiangsu Collaborative Innovation Center of Chinese Medicinal Resources Industrialization, National and Local Collaborative Engineering Center of Chinese Medicinal Resources Industrialization and Formulae Innovative Medicine, Nanjing University of Chinese Medicine, Nanjing 210023, China; 20181360@njucm.edu.cn (M.G.); scotter01@163.com (B.B.); raindc@163.com (Y.C.); shanmingqiu@163.com (M.S.); cff19870524@163.com (F.C.); jiangmiao820701@163.com (M.J.)

**Keywords:** pollen grains of Typha, thermal analysis, carbonizing process, Fourier transform infrared spectrometry (FTIR)

## Abstract

Carbonized pollen grains of Typha (CPT) were widely used in clinical for antithrombosis, wound and bleeding in China. In order to ensure the role of drugs, it is very important to control the quality of drugs. However, there is a lack of monitoring methods in the process of charcoal preparation. To characterize the process of CPT, we used thermal analysis, scanning electron microscope (SEM), color measurement, Fourier transform infrared spectrometry (FTIR) and HPLC. In this study, 7 min was the optimal processing time and the heating process condition should be controlled under 272.35 ± 7.23 °C. This comprehensive strategy to depict the whole carbonizing process would provide new ideas for researches on quality control of Traditional Chinese Medicine (TCM) and processing theory of charcoal medicine.

## 1. Introduction

Charcoal medicine, named ‘Tan Yao’ in China, is a kind of special processing product in Chinese medicine and used for treatment of hemoptysis, hematemesis, and hemorrhage in clinic during ancient times [1]. The principal way to prepare charcoal medicine is carbonizing the drugs by high-temperature heating [2]. After being charred, the chemical compositions in drugs will be changed [3]. Some of the components will be lost during the process of carbonizing but the efficacy of hemostatic will be enhanced and the toxic effect of the crude drug will be reduced according to the Traditional Chinese Medicine (TCM) theory [4]. 

Pollen *Typha* (PT) is a well-known TCM medicine widely used in clinical for antithrombosis [5], wound, and bleeding [6]. Both the raw pollen grains and the PT charcoal are listed in the Pharmacopoeia of the People’s Republic of China (2015 edition). There are two processing methods to prepare PT charcoal. One is putting the pollen grains into a heating container to stir fry them to deep yellow, named stir frying PT (SPT) and the other is baking them to brown, named carbonized PT (CPT) [7]. Previous researches have demonstrated that the anti-hemorrhagic activity of PT was enhanced after carbonized processing [8]. Moreover, compounds in the different process products showed significant differences due to the heating time. The flavonoid glycosides were gradually decreasing while aglycones content, such as quercetin, kaempferol, and isorhamnetin, were increased. Meanwhile, some unknown compounds appeared in CPT [9,10]. However, the mechanism of chemical composition transformation and hemostatic effect of CPT are still unclear. Furthermore, there are no strict control measures and corresponding quality standard for CPT during the heating process. 

Thermal analysis is a method to measure the physical or chemical properties by monitoring against time or temperature during the process of heating [11]. Thermal analysis has been used as a method for evaluating the process technology of Chinese medicine in recent years [12]. This technology could provide various data to researchers including thermal gravimetry (TG), differential thermal analysis (DTA), derivative thermogravimetry (DTG), dynamic mechanical analysis (DMA), and thermo-mechanical analysis (TMA) for studying the changes of drugs during frying [13]. FTIR is one of the widely used methods for elucidating and identifying the components in herb medicine, and providing the date for the properties of the constituents in TCM [14,15].

Thus, in order to provide basis for further optimization of PT process technology, the purposes of the present study are: (1) preparing CPT products and analyzing the changes in appearance and color of samples during the processes and (2) analyzing the characteristics of PT during frying processes by thermal analysis, FTIR and the changes of different constituents. 

## 2. Results and Discussion

### 2.1. Pollen Grains Observed by SEM during Heating Process

Pollen grains of PT were free, isopolar, and spherical in equatorial and polar. A small amount of male inflorescence residue was dangled around the pollen grains. The residue turned to circular with smooth margin during the heating process. The pollen grains of CPT were obviously shrunken and the reticulate ornamentation was not uniform size with slightly decreasing after heating. At last, the pollen grains were cracked and the reticulate ornamentation was stick to each other and carbonized after continuous heating as shown in Figure 1.

The pollen grains classification of *Typha* was based on the morphological characters of configuration of the surface of pollen such as the hole in the germinal aperture and the width of the reticulate ornamentation [16,17]. Moreover, the pollen of *Typha* is classified into two types, one is tetrads in which the pollen grains are released in clusters of four such as *T. latifolia* and *T. angustifolia*, the other is monads in which the pollen grains occurs in single form such as *T. angustifolia* and *T. orientalis* [18]. The pollen in our experiment was consistent with the species *T. orientalis* Presl as described in the literature. During the heating process, the pollen grains exposed to the heating vessel were carbonized and the reticulate ornamentation were contacted and adhered together. The spherical surface of the pollen grains was not changed significantly until heated for 5 min, and most of the grains were shrunken at 7 min. Some of the pollen grains were cracked after continuous heating and carbonized after heated for 10 min. 

### 2.2. Color Measurement

The parameters of PT in this experiment were shown in Table 1. The b* value of PT was 45.96, but gradually decreased during the stir frying process and reached the lowest value 4.19 after being heated for 10 min. Contrary to b* values, the a* values gradually increased until heated for 5.5 min from 5.42 to 10.59 and progressively decreased. As a result, the color of PT was changed from yellow to black. 

The reflectivity of the sample absorption wavelength during the heating process was measured as shown in Figure 2A. The value of PT reflectivity was observed at approximately 0.75 and gradually decreased after fried process. The values declined relatively mild initially and less than zero after heated for 4 min. The reflectivity values dropped suddenly at about 5 min and approached −0.5 after 7 min. The transmittance was close to a straight line after continually heating. Furthermore, we reproduced the color changes during the process using the values of accurate L*a*b* color units as shown in Figure 2B.

The aspect of the herb medicine was one of the important parameters to evaluate the TCM process technology. Many methods based on color measurement have been developed for herb medicines evaluation [19,20]. The most used model to describe color was RGB model which can elucidate the intensity of light in red, green, and blue, and the values of accurate L*a*b* color units can be transformed from RGB model [21]. Color is an important quality control index of CPT in the Pharmacopoeia of the People’s Republic of China (2015 edition), in which the regulated color of processed pollen charcoal is dark brown. In this experiment, the color of different processed products and the digital color values were measured. The color of PT was yellow and the CPT was brownish black, the pollen grains tuned black after completely carbonized after heated for 10 min. 

### 2.3. FTIR Diversity of PT

The samples of CPT in the frying process were analyzed by FTIR as shown in Figure 3.

There are four main bands in infrared spectroscopy located at band a (3696–3004 cm^−1^), band b (3011–2833 cm^−1^), band c (1744–1550 cm^−1^), and band d (1195–975 cm^−1^), respectively. The strong and broad absorption peak at band a was the stretching vibration of ν_O-H_, which was due to the alcohol hydroxyl or phenolic hydroxyl groups in the compounds of PT [22]. The wider band was because of the formation of hydroxyl intramolecular hydrogen bonds [23]. Furthermore, the bands a were strong in the samples of PT after heated for 2 min and 7 min, but weakened after heated for 10 min indicating that the compounds were decomposed under this heating conditions [24]. The peak was observed at region b assigned to the stretching vibration of ν _C-H_ in saturated hydrocarbon [25]. All the samples of PT have strong absorption peaks in region b. Bands c were assigned to the stretching vibration of ν_C=C_ and ν_C=O_ [26,27]. The absorption peak was shift toward lower waves at 1648 cm^−1^ because the carbon-carbon double bond conjugated with carbonyl groups [28]. In this experiment, the two bands merged into a single peak, but gradually bifurcated with the temperature increasing. This result indicated that the compounds in PT were gradually decomposed and the conjugated double bonds and carbonyls decreased. The peak was observed at band d was assigned as the stretching vibration of ν_C-O_ [26]. 

In order to study the infrared spectra of the compounds in PT during heating process, we studied the ratio of band a to band b, c, and d as shown in Figure 3B. The ratio of band a to b had been decreasing, and there were two significant drops at samples heated for 4 min and 9 min. PT contains flavonoid glycosides, polysaccharides, and alkanes which have hydroxyl groups. Experimental results demonstrated that the hydroxyl groups of the compounds in PT gradually decreased, especially at the fourth and ninth minutes in the heating process, while the structure of alkanes were relatively stable. The ratio of band a to c also reduced in the sample heated for 7 min, then increased and reached the highest values at 9 min, finally, the ratio dropped rapidly. It can be concluded that when the temperature reaches a certain value, some of the compounds which contain double-bond may decompose. The ratio a:d was continuously rising until the sample heated for 7 min, and then a sudden fall to near zero. This phenomenon suggested that the ether bond had been slowly decomposed before the decomposition of the hydroxyl compound during the heating processes.

### 2.4. Thermal Analysis

Temperature control is one of the important parameters during herb medicine heating process. In order to elucidate the changes of properties of CPT, we analyzed the thermal and the thermodynamics of the samples during the pollen grains frying.

#### 2.4.1. Thermogravimetric Analysis of PT 

Thermal analysis is a group of techniques to study the physical and chemical of properties during heating process in atmosphere [11]. Hemicellulose, cellulose and lignin are usually considered as the major compounds changed in the heating processes [29,30]. In the typical TG-DTG curves of natural material, the first peak indicates that primary pyrolysis reaction begins when the temperature is close to 150 °C and a large number of volatiles and gases will be produced. Then the second and third peaks will be observed at higher temperature as a result of decomposition of hemicellulose and cellulose [31].

Figure 4 depicted the TG and DTG curves of PT at the heating rate of 5 °C min^−1^, 10 °C min^−1^, 15 °C min^−1^, and 20 °C min^−1^. As shown in Figure 4A, weight loss accompanied by the increasing heating rates indicted that thermal decomposition occurred slowly at higher temperatures [32]. This phenomenon may be caused by the aggregation of pollen grains after heating. 

In Figure 4B, peak 1 located at about 150 °C was obviously observed as a “shoulder”. This peak was usually considered as the removal of water. There was a major mass loss appearing in the temperature between 200 °C and 450 °C, during which most of the volatiles were released. This phenomenon indicated that compounds in pollen grains decomposed into small molecular compounds because of the continuous heating. The mass conversion rates in the temperature from 200 to 450 °C were 74.13% (5 °C min^−1^), 71.72% (10 °C min^−1^), 66.87% (15 °C min^−1^), and 65.76% (20 °C min^−1^), respectively. Previous researches have demonstrated that the lower heating rate, the more compounds in the materials combusted and pyrolyzed because the constituents in herb medicine have sufficient time to react [13]. The peak 2 and peak 3 were known as oxidative pyrolysis. Peak 2 in the DTG experimental curve was considered corresponding to the decomposition rate of the hemicellulose representing the volatile matter accumulated products [33]., The main characteristic of PT was peak 2 in this experiment, and peak 4 was caused by carbonization of substance char combustion involved in the PT’s frying process. 

The thermal parameters values including the pyrolysis starting temperature (T*_v_*), the temperature for maximum weight loss (T*_m_*), the final temperature for pyrolysis (T*_f_*), and the maximum weight loss rate (DTG*_max_*) were observed in our study. The characteristic parameters were showed in Table 2.

There was a trend to delay thermal decomposition process towards higher temperature range when the heating rate increased. The beginning and final temperatures of breakdown for hemicellulose and cellulose increased, which also signified the shift of devolatilization and peak temperatures. 

The peak of maximum thermal weight loss was at 293.5 ± 8.54 °C. and the pyrolysis finished at 364 ± 11.69 °C. The data indicated that the temperature should be controlled during the PT heating process based on the parameters coming from pyrolysis characteristics. The chemical composition in pollen grains will be decomposed at a certain temperature.

#### 2.4.2. Kinetic Analysis Using Iso-Conversional Models.

The iso-conversional model has been wildly used in kinetic analysis [34]. This model can avoid the system errors caused by Arrhenius parameter estimation [35]. In the present work, we used Kissinger–Akahira Sunose (KAS), Ozawa–Flynn–Wall (OFW), and Friedman based on the iso-conversional model to describe the heating process of PT. The activation energies of various conversions can be calculated by the slopes of liner relationships. Although some liner correlation coefficients (R^2^) were small, we still took them into consideration because the thermal decomposition was a constant process in our study. The activation energies were calculated using the three different models and all specific parameters were listed in Table 3.

The plots of the Friedman, OFW, and KAS methods showed the trend of activation energy. The conversion (α) in Table 4 was divided into two stages, the first stage (0.05–0.35) which represented the status of pyrolysis and the second stage (0.35–0.9) which featured the status combustion [36]. There were two characteristics in our experiment. One was the thermo-oxidative degradation in stage 1 which had higher activation energies than the second stage and the calculated activation energies in the three methods were practically equal. These different activation energies reflected the thermodynamic equilibrium in pollen grains. The other characteristic phenomenon was that the highest activation energy of PT appeared at the conversion rate of 0.30. The highest activation energy was 168.57 kJ·mol^−1^, 175.18 kJ·mol^−1^, and 174.94 kJ·mol^−1^, respectively. When the conversion rate reached 0.30, the temperature was 272.35 ± 7.23 °C. The date was consistent with the temperature of maximum thermal weight loss. These results suggested that the temperature should be controlled under 272.35 ± 7.23 °C during the heating process.

The activation energies were estimated based on the KAS, OFW, and Friedman models for PT as shown in Figure 5. 

Previous research had demonstrated that pollen wall consisted of two phases which contained compounds including cellulose and other polysaccharides [37]. The activation energy of this kind of materials can be calculated from the models Friedman, KAS, and OFW based on the equation discussed in section 3, Table 5. As shown in Figure 5A–C, the straight lines were formed depending on the degree of conversion α. The states of activation energy lines implied different reaction mechanisms, the parallel activation energy lines indicated that a similar reaction mechanism involved in the heating process of pollen grains [30,36]. The changes between comparatively higher conversion periods and lower periods (α < 0.2) indicated the possible different reaction mechanisms. As shown in Figure 5D, there was no significant difference in the three models indicating that the active energy data obtained from the models were reliable and effective.

### 2.5. Flavonoids Content Changes during Heating Process

Flavonoids are wildly distributed in PT. Typhaneoside and isorhamnetin-3-O-neohespeidoside are the two main constituents which are the ingredients of the quality control for the pollen in the Pharmacopoeia of the People’s Republic of China (2015 edition). The same aglycone of the two flavonoid glycosides is isorhamnetin. During the process of heating, the content of glycosides was gradually decreasing, but the content of aglycone was gradually increased. Furthermore, some new chemical components will be produced [9,10]. The content of typhaneoside, isorhamnetin-3-*O*-neohespeidoside and isorhamnetin were determined in experiment. The determination of recovery test was showed in Table 4. The HPLC chromatography of different samples from 1 min to 10 min were showed in Figure 6A, and the ratio of compounds content in pollen grains compared with PT were showed in Figure 6B. After heated for 7 min, the content of isorhamnetin in the samples reached its maximum value but the glycosides almost all decomposed. The data observed from the study was consisted with our previous research that the optimum condition for preparing CPT was heated for 8 min at 210 °C [38]. 

## 3. Materials and Methods

### 3.1. Materials and Reagents

Pollen grains of Typha was collected from Tongzhou (Jiangsu, China), the voucher specimen number was 20180820. The raw pollen grains were identified as the pollen grains of *T. orientalis* Presl by vice Prof. Yan Hui, department of medicinal plants, Nanjing University of Chinese Medicine and deposited at the laboratory for chemistry of Chinese medicine. Typhaneoside (Lot: Y14S8H43976), isorhamnetin-3-*O*-neohespeidoside (Lot: H30M3X1) and isorhamnetin (Lot: P23A9F68614) were all bought from Shanghai Yuanye Biological Technology Co., Ltd. (Shanghai, China). Glacial acetic acid (Lot: 1503102) was purchased from Shanghai Shenbo Chemical Co., Ltd. (Shanghai, China). Ethanol (Lot: 20180715) came from Wuxi City Yasheng Chemical Co., Ltd. (Wuxi, China). Isoamyl acetate solution (Lot: l1706040) and phosphoric acid (Lot: C1801089) were obtained from Aladdin Industrial Corporation (Shanghai, China). Furthermore, methanol was from Jiangsu Hanbon Sci. Tech. Co., Ltd. (Jiangsu, China). Glycerol and KBr were provided by Sinopharm Chemical Reagent Co., Ltd. (Shanghai, China). Acetonitrile obtained from Merck (Darmstadt, Germany).

### 3.2. Sample Preparation

400 g pollen grains were heated in a stainless steel pan on an electromagnetic furnace (Midea, Guangdong, China) at 200 °C, with continuously stirring during heating, and took 20 g grains out from the pan every half a minute until 10 min. 

### 3.3. Pollen Grains Observed by SEM during Heating Process

The method of preparing pollen grains samples for SEM was modified from literature [39]. Pollen grains were immobilized in FAA fixative solution (5 mL 38% formaldehyde, 5 mL glacial acetic acid, 90 mL 50% alcohol and 5 mL glycerol) for 24 h at 4 °C. The samples were dehydrated successively by ethanol (70%, 80%, 90%, 95%, and 100% for 20 min, respectively) at 4 °C. Then the pollen grains were treated with mixed ethanol and isoamyl acetate solution (75%:25%, 50%:50%, 25%:75%, and 100% isoamyl acetate for 20 min, respectively) before freeze drying for 48 h. 

Pollen grains in heating processes were studied through SEM (Hitachi S-4800, Tokyo, Japan). Samples were examined on 8.4 mm diameter stubs with sticky tabs, coated in a sputter coater with approximately 25 μm of Gold Palladium. The samples were photographed at an accelerating voltage of 5.0 KV.

### 3.4. Color Measurement

The color measurement was analyzed by CM3600A spectrophotometer (Konica Minolta, Tokyo, Japan), multifunctional infrared oven (Guangdong Beauty Electric Appliance Manufacturing, Guangdong, China). A PMPLUS Infrared thermometer (Raytek, California, USA) was used for determined the temperature. 

The light source was pulsed xenon lamp system including specular reflection (Specular component included, SCI) and exclude specular reflection (Specular component excluded, SCE) with a standard viewing angle at 8 degrees. The measuring diameter was set at 3 mm, measuring wavelength was set at 360~740 nm. The standard deviation between instruments (ΔE*_a*b*_) was 0.15. ΔE* (color difference) was calculated as follows: ΔE* = {(ΔL*)^2^ + (Δa*)^2^ + (Δb*)^2^}^1/2^. ΔL* was the brightness of specimens, calculated as follows: ΔL* = value L* of specimens in the conventional layering system (CLS) group − value L* of specimens in digital veneering system (DVS) group; Δa* indicated the degree of red-green samples and Δa* = value a* of CLS − value a* of DVS; Δb* was the degree of yellow-blue samples and Δb* = value b* of the CLS − value b* in the DVS [40]. The parameters for determination color were L*, a* and b*. L* indicated brightness from black to white, a* and b* axis indicated the changing color. The positive direction of the a* axis represented the color to red, and the negative direction represented the color to green; the positive direction of b* axis represented the color to yellow and the negative direction represented to blue. The standard value was set as the corrected whiteboard values (98.84, −0.29 and 0.23).

### 3.5. FTIR Diversity of PT 

FTIR spectrum of all samples were measured by FTIR spectrometer (Thermo Scientific Nicolet iS5, Waltham, MA). Each sample was mixed at a ratio of 1:100 with KBr, and then detected in the range of 4000~400 cm^−1^. The spectral resolution was set at 4 cm^−1^, and the scan signal was accumulated 16 times. The external influence such as H_2_O and CO_2_ was removed automatically. Automatic baseline correction and curve smoothing were performed on the software Thermo Fisher Nicolet FTIR OMNIC 8.0.

### 3.6. Thermogravimetric Analysis

Approximately 3 mg of samples were determined at a thermogravimetric analyzer (Dimond TG/DTA, PE, Waltham, MA, American). The samples were placed in a platinum pan, the column was heated to raise the temperature from 25 to 600 °C and the reactive gases with the flow rate of nitrogen 15 and 100 mL·min^−1^, respectively. Experimental runs were carried out at four different heating rates (5 °C·min^−1^, 10 °C·min^−1^, 15 °C·min^−1^ and 20 °C·min^−1^, respectively) to determine the pyrolysis characteristics. The kinetic parameter analysis was based on the data obtained from thermogravimetric analysis (TGA) experiment. 

#### Mathematical Background

The thermodynamic properties of materials are important for kinetic study. In the analysis, only three to five different heating rates obtained from TG and different kinetic analysis methods can provide believable kinetic study result [41]. Pyrolysis of nature biomass is too complex to understand during the thermal degradation, as a result, the exact reaction mechanism of biomass is impossible to demonstrate. However, many studies have been described the pyrolysis process by simplified model [31]. 

The pyrolysis reaction mechanism proposed by Sadhukhan et al. [42] was described as follows: (1)$$\mathrm{Biomass}\;(\mathrm{Solid})\overset{k\left(t\right)}{\to }\mathrm{volatile}\;(\mathrm{gases}\;+\;\mathrm{tar})\;+\;\mathrm{Char}\;(\mathrm{Solid}\;\mathrm{residue})$$
where *k* is the rate constant of the reaction. Generally, the kinetic analysis can be described as follows [43]:(2)$$\frac{d_{\alpha }}{d_{t}}=\mathrm{kf}\left(\alpha \right).$$

The Equation (2) expresses the reaction rate of conversion at a constant temperature in the reaction model f(*α*), α can be expressed through the formula:(3)$$\alpha =\frac{m_{i}-m_{t}}{m_{i}-m_{f}}.$$
where m*_i_* is the initial weights, m*_f_* is the final weights and m*_t_* is the current weights of samples at the moment *t*. According to Arrhenius equation [43]:(4)$$k\left(T\right)=\mathrm{Aexp}\left(\frac{-E\alpha }{\mathrm{RT}}\right)$$
where T = absolute temperature (K), A = pre-exponential factor, Eα = the activation energy (kJ·mol^−1^), R = gas constant (8.314 J·mol^−1^·K^−1^), k(T) is the temperature-dependent reaction rate. In our experiment, the IPR model was used to analysis the primary conversion of pollen grains into gases and volatiles. The devolatilization can be expressed by [43,44]. Substitution Equation (4) in Equation (2) gives:(5)$$\frac{d_{\alpha }}{d_{t}}=\mathrm{Aexp}\left(\frac{-E\alpha }{\mathrm{RT}}\right)f\left(\alpha \right).$$

Based on the Arrhenius equation, introducing β = dT/dt as the heating rate into Equation (5), a typical gas solid reaction can be described as follows [36]:(6)$$\frac{d_{\alpha }}{d_{T}}=\left(\frac{A}{\beta }\right)\exp\left(\frac{-E\alpha }{\mathrm{RT}}\right)f\left(\alpha \right).$$

The common models used in TGA, the Friedman model, Kissinger-Akahira-Sunose (KAS) model and Flynn-Wall-Ozawa (FWO) model, were listed in Table 5. Friedman model is conditioned by an invariance of E versus α, FWO method and KAS method plots of lnq versus 1/T and ln(q/T) versus 1/T, which can calculate the apparent activation energy values during the thermal degradation at different values of α [35,44].

### 3.7. Determination of the Compounds during Heating Process

The HPLC analysis used Agilent 1290 (Agilent Technologies Inc, Santa Clara, CA, USA). Pollen grains of Typha were soaked with methanol (1:20 w/v) overnight and refluxed for 1 h, respectively. The extracted solution was subject to a chromatography PRAZIS Absolute C18 column (250 mm × 4.6 mm, 5 μm) (Shanghai Dahu Scientific Instrument Co., Ltd., Shanghai, China). The mobile phase was water with 0.05% phosphoric acid (A)-acetonitrile (B) with gradient elution, and the gradient program was showed in Table 6. The flow rate was set at 1 mL·min^−1^ and the injection volume was 10 μL.

To develop the calibration curves, a range of different concentrations of three standard solutions were configured and detected under the above conditions. Table 7 showed the linear calibration curves, r^2^, the LOD and LOQ of typhaneoside, isorhamnetin-3-*O*-neohespeidoside and isorhamnetin. The data obtained from the present HPLC method were considered to be satisfactory for subsequent analysis of all of the samples.

## 4. Conclusions

In the present work, we studied the appearance, color, FTIR, thermal characteristic, and the content changes of compounds in PT during the heating process. After being heated for more than 7 min, the pollen grains were shrunken and broken, the color turned to black. The absorption peak area which represent hydroxyl group reduced sharply in the infrared spectra indicating some compounds had been decomposed.

The contents of flavonoids glycosides decreased during the heating and their aglycones increased until after heated for 7 min. The heating rate, heating time and heating temperature were all the important effects on the process for preparing CPT and the heating process condition should be controlled under 272.35 ± 7.23 °C Our present work would provide new ideas and methods for charcoal medicine preparation.

## Figures and Tables

**Figure 1 molecules-24-00128-f001:**
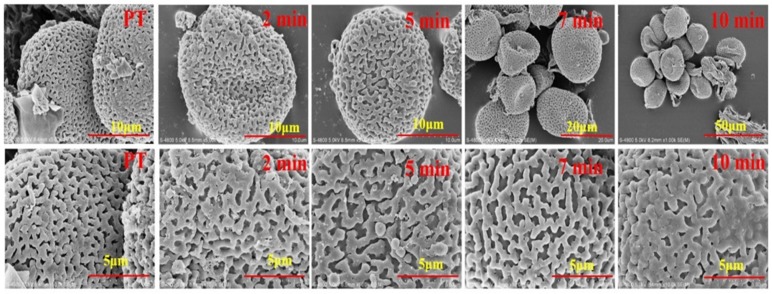
Scanning Electron Microscope of pollen *Typha* (PT) samples during frying processes.

**Figure 2 molecules-24-00128-f002:**
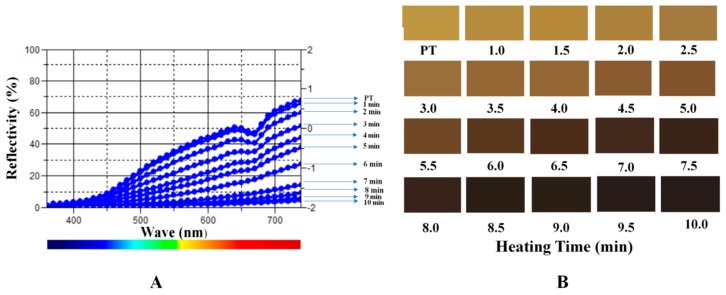
The reflectivity of the PT in heating process (**A**) and the colors of samples (**B**).

**Figure 3 molecules-24-00128-f003:**
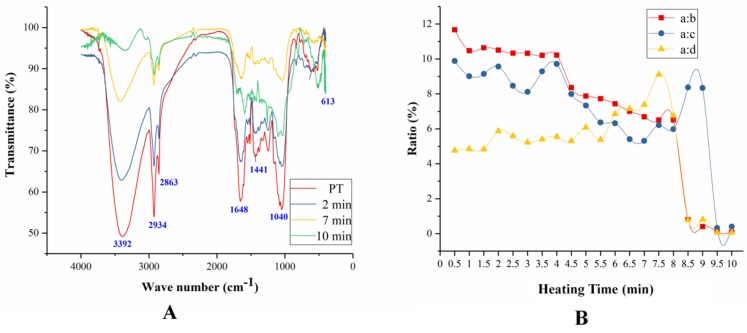
The FTIR of PT samples during the frying processes (**A**) and the ratio of band a to band b, c, and d (**B**).

**Figure 4 molecules-24-00128-f004:**
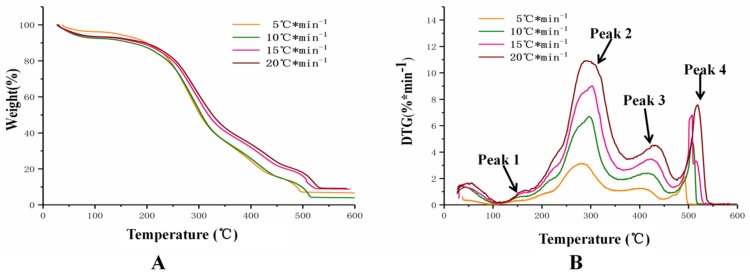
The TG-DTG curves of PT at four different heating rates. Notes: (**A**) was TG curves and (**B**) was DTG curves.

**Figure 5 molecules-24-00128-f005:**
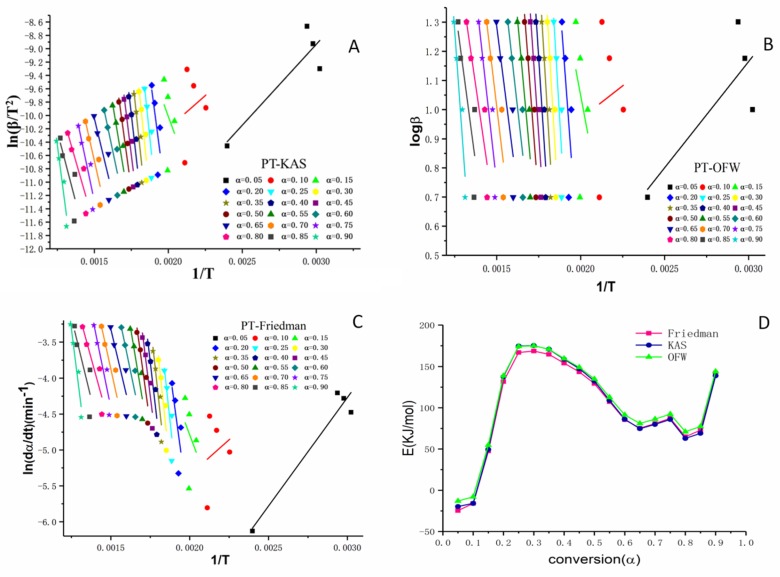
Estimation of activation energy using the KAS, OFW, and Friedman models for PT. Notes: Linear relationship curves (**A**–**C**) were made according to KAS, OFW and Friedman models at different conversion rates (from 0.05 to 0.90), respectively; (**D**) was the curve fitting charts of the three models.

**Figure 6 molecules-24-00128-f006:**
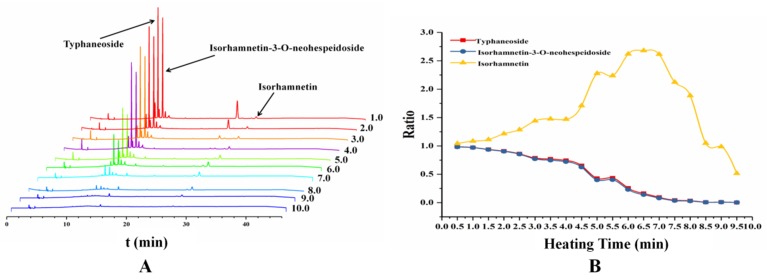
The content changes of the two main flavonoid glycosides and its aglycone in different samples heated from 1 to 10 min. (**A**) the HPLC chromatography of different samples from heating for 1 min to 10 min and (**B**) the ratio of compounds content in pollen grains compared with PT.

**Table 1 molecules-24-00128-t001:** The values of color measurement of PT.

Sample	L*	a*	b*	dL*	da*	db*	dE*ab
PT	65.26	5.42	45.96	−9.77	3.85	27.09	29.06
0.5 min	64.19	5.55	44.91	−10.83	3.98	26.04	28.48
1.0 min	64.19	5.68	44.45	−10.84	4.1	25.58	28.09
1.5 min	61.1	6.36	42.8	−13.93	4.78	23.93	28.1
2.0 min	59.76	6.93	42.53	−15.26	5.36	23.66	28.66
2.5 min	57.19	7.64	39.26	−17.83	6.07	20.39	27.76
3.0 min	53.93	8.09	35.31	−21.09	6.52	16.44	27.53
3.5 min	49.86	9.08	32.18	−25.17	7.51	13.32	29.44
4.0 min	47.3	10.03	30.67	−27.73	8.46	11.8	31.3
4.5 min	47.12	10.22	30.69	−27.9	8.65	11.82	31.51
5.0 min	42.18	10.57	27.95	−32.84	9.0	9.08	35.25
5.5 min	38.54	10.59	25.54	−36.48	9.02	6.67	38.17
6.0 min	34.32	10.35	23.18	−40.7	8.78	4.31	41.86
6.5 min	26.86	9.03	17.24	−48.16	7.45	−1.63	48.76
7.0 min	23.28	8.09	14.21	−51.74	6.51	−4.66	52.36
7.5 min	21.48	7.0	11.28	−53.54	5.43	−7.59	54.35
8.0 min	18.37	5.88	8.97	−56.65	4.31	−9.9	57.67
8.5 min	18.24	5.61	8.25	−56.79	4.03	10.62	57.91
9.0 min	15.25	4.25	6.21	−59.77	2.68	12.66	61.16
9.5 min	15.08	4.26	6.16	−59.94	2.69	−12.7	61.34
10.0 min	13.89	3.15	4.19	−61.13	1.58	14.68	62.89

Notes: L*: the brightness of the color; a*: the degree of red or green; b*: the degree of yellow or blue.

**Table 2 molecules-24-00128-t002:** Parameters of pyrolysis characteristics of PT.

Sample	β (°C·min^−1^)	T*_v_* (°C)	T*_m_* (°C)	T*_f_* (°C)	DTG*_max_* (%·min^−1^)	Volatiles (%)
PT	5	150	283	348	3.15	39.30
	10	141	297	365	6.71	45.25
	15	170	303	367	9.03	42.90
	20	172	291	376	10.93	34.55

**Table 3 molecules-24-00128-t003:** Parameters of PT calculated by Kissinger–Akahira Sunose (KAS), Ozawa–Flynn–Wall (OFW), and Friedman.

Conversion (α)	Activation Energy Friedman model (kJ·mol^−1^)	R^2^	Activation EnergyKAS Model (kJ·mol^−1^)	R^2^	Activation Energy OFW Model (kJ·mol^−1^)	R^2^
0.05	−24.58	0.9419	−19.72	0.8059	−12.98	0.6691
0.10	−15.96	0.0500	−15.87	0.0417	−8.01	0.0121
0.15	47.60	0.0980	49.48	0.0916	54.72	0.1202
0.20	131.72	0.5184	137.59	0.4983	138.83	0.5282
0.25	166.99	0.7681	174.65	0.7437	174.26	0.7618
0.30	168.57	0.8300	175.18	0.8116	174.94	0.8263
0.35	164.68	0.8604	170.79	0.8442	170.92	0.8574
0.40	154.01	0.8721	158.73	0.8555	159.62	0.8690
0.45	143.38	0.8730	147.36	0.8581	148.96	0.8726
0.50	129.49	0.8563	132.44	0.8420	134.95	0.8598
0.55	107.32	0.8112	108.87	0.7924	112.75	0.8194
0.60	85.59	0.7768	86.11	0.7518	91.42	0.7908
0.65	75.21	0.7814	74.72	0.7507	80.98	0.7966
0.70	80.95	0.8272	79.88	0.7965	86.28	0.8350
0.75	87.31	0.8650	85.88	0.8372	92.34	0.8683
0.80	64.88	0.8353	62.91	0.8009	70.97	0.8503
0.85	72.88	0.7266	69.07	0.6774	77.34	0.7444
0.90	142.02	0.9338	138.81	0.9317	144.01	0.9267

**Table 4 molecules-24-00128-t004:** Recovery of the three constituents.

Compound	Content (mg)	Detected (mg)	Added (mg)	Recovery (%)	Average Recovery (%)	RSD (%)
Typhaneoside	0.1806	0.3601	0.1828	98.21	96.67	1.70
	0.1841	0.3609	0.1828	96.74		
	0.1822	0.3534	0.1828	93.65		
	0.1832	0.3630	0.1828	98.37		
	0.1818	0.3581	0.1828	96.43		
	0.1810	0.3576	0.1828	96.63		
Isorhamnetin-3-*O*-neohespeidoside	0.1466	0.2931	0.1445	101.40	101.84	1.77
	0.1495	0.2963	0.1445	101.57		
	0.1443	0.2897	0.1445	100.62		
	0.1451	0.2946	0.1445	103.47		
	0.1440	0.2879	0.1445	99.63		
	0.1433	0.2941	0.1445	104.35		
Isorhamnetin	0.002949	0.005846	0.002846	101.81	100.88	1.86
	0.002903	0.005770	0.002846	100.73		
	0.002931	0.005698	0.002846	97.25		
	0.002953	0.005842	0.002846	101.53		
	0.002868	0.005782	0.002846	102.39		
	0.002886	0.005777	0.002846	101.57		

**Table 5 molecules-24-00128-t005:** Kinetic methods used in evaluating activation energy in the experiment.

Method	Expression	Plots
Friedman	$\ln\left(d_{\alpha }/d_{t}\right)=\ln\left[\mathrm{Af}\left(\alpha \right)\right]-E_{\alpha }/\mathrm{RT}$	$\ln\left(d_{\alpha }/d_{t}\right)$ against 1/T
KAS	$\ln\left(\beta /T^{2}\right)=\ln\left(\mathrm{AR}/E_{\alpha }\right)+\left(1/T\right)\left(-E_{\alpha }/R\right)$	$\ln\left(\beta /T^{2}\right)$ against 1/T
FWO	$\log\beta =\log\left({\mathrm{AE}}_{\alpha }/\mathrm{Rg}\left(\alpha \right)\right)-2.315-0.4567E_{\alpha }/\mathrm{RT}$	logβ against 1/T

**Table 6 molecules-24-00128-t006:** The gradient elution.

Time(min)	A (%)	B (%)
0.00	95.00	5.00
5.00	86.00	14.00
10.00	68.50	31.50
15.00	68.50	31.50
40.00	55.00	45.00
43.00	95.00	5.00
46.00	95.00	5.00

**Table 7 molecules-24-00128-t007:** Calibration curves, limits of detection (LOD), and limits of quantitation (LOQ) of the three analytes.

Component	Calibration Curves	r^2^	Linear (μg)	LOQ (ng)	LOD (ng)
Typhaneoside	y = 11665x − 3.0371	0.9997	0.007516–0.4810	0.6413	0.4276
Isorhamnetin-3-*O*-neohespeidoside	y = 14729x − 2.4082	0.9997	0.004838–0.3096	1.032	0.4128
Isorhamnetin	y = 33789x − 1.1673	0.9997	0.002021–0.03234	1.0870	0.3623

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
