# Peer review of "Chemical Property Changes and Thermal Analysis during the Carbonizing Process of the Pollen Grains of Typha"

_molecules, 2018, doi:10.3390/molecules24010128_

Round 1

Reviewer 1 Report

This paper describes the chemical property changes and thermal analysis during the carbonizing process of the pollen grains of typha. The article is quite complete, it is of interest to the scientific community, the methods used are appropriate and the statistics used are correct. The work is interesting and deepens in the knowledge on the carbonizing process of the pollen grains of typha. On the other hand the English and the expressions need to be corrected by an expert English reviewer. The article must be written with expressions and phrases in a more scientific format. Many other issues should be improved before being published.

I consider that the article is not appropriate to be published in Molecules journal in this form. Authors have to make major modifications to it before it will be published.

Other observations:

First of all, extensive corrections of English.

The authors abuse the first person plural (we, our). Write in a more scientific format.

The format of the article must be completely revised according to the format of the Molecules journal. Use the template of the journal.

Title: Each word should be capitalized.

Lines 60, 140, 145, …….:  Capitalized each word according the specification of the journal.

Lines 20, 205, 207,……: Put separations after and before “±”. Apply to the entire document.

Line 23: Delete the separation after “Typha”.

Lines 43, 54, ….: Put the references in the journal format. For example, [9,10] instead of [9-10]. Apply to the entire document.

Line 60: The title of the section is too long.

Line 60: “2.1.” instead of “2.1”. Apply to the entire document.

Lines 66, 68…..: Use Figure instead of Fig. according with the format of the journal. Apply to the entire document.

Line 81: Capitalized Table.

Figure 3: Increase the resolution of the figure.

Line 144: Delete

Figure 4: Axis: put separation before ( ).

Lines 153, 154, …….: Revise the format of ºC. Unify and apply to the entire document.

Table 2: Unify the format of tables. Delete most of the lines for separations. Volatiles instead of volatlles.

Line 230: The title of the section is too long.

Figure 6: Unify the x axis: Time or t.

Line 254: Capitalize Hui.

Lines 256, 261: Do not capitalize isorhamnetin and methanol.

Lines 265, 266: Put a separation between the number and g.

Line 268: The title of the section is too long.

Sections 3.3, 3.4, …..: For each equipment add brand, city and country. Unify and apply to the entire document.

Line 273: Delete the separation after “:”. Unify.

Lines 287, 290, …….: Put separations after and before “=”. Apply to the entire document.

Section 3.7: Describe the compounds analyzed by HPLC. Indicate the range of each compound to prepare the calibration curve. Indicate r2, LOD, LOQ….

Line 343: ¿American?. USA.

Line 346: water with 0.05% phosphoric acid.

Line 345: Indicate the brand, city and country of the column.

References: Use the format of the journal.

Author Response

Dear Editor:

Thank you very much for your attention and the referee’s evaluation and comments on our manuscript entitled “Chemical Property Changes and Thermal Analysis during the Carbonizing Process of the Pollen Grains of Typha” (Manuscript ID: molecules-410664). We would like to express our sincere gratitude to the reviewer’s kindness and patience on reviewing our manuscript. We have now carefully revised our manuscript according to the reviewers’ comments and suggestions. A revised manuscript with the correction sections red marked was attached as the supplemental material and for easy check purpose. Should you have any questions, please contact us without hesitate.

Best regards,

Mingliang Gao

Name: Peidong Chen & Li Zhang

E-mail address: cpd@njucm.edu.cn (P.C.); zhangli@njucm.edu.cn (L.Z.)

2018-12-23

(1) Extensive corrections of English

Response: Thanks for your suggestion. We have invited an expert English reviewer to correct the English and the expression.

(2) The authors abuse the first person plural (we, our). Write in a more scientific format.

Response: Thanks for your suggestion. We have revised in the manuscript.

(3) Title: Each word should be capitalized.

Response: Sorry for this mistake. We have capitalized each word of the title.

(4) Lines 60, 140, 145, …….: Capitalized each word according the specification of the journal.

Response: We have capitalized each word according the specification of the journal.

(5) Lines 20, 205, 207,……: Put separations after and before “±”. Apply to the entire document.

Response: Sorry for these mistakes. We have put separations after and before “±”.

(6) Line 23: Delete the separation after “Typha”.

Response: Sorry for this mistake. We have deleted the separation after “Typha”.

(7) Lines 43, 54, ….: Put the references in the journal format. For example, [9,10] instead of [9-10]. Apply to the entire document.

Response: Sorry for these mistakes. We have put the references in the journal format.

(8) Line 60: The title of the section is too long.

Response: We have changed the title of this section according to the reviewer’s suggestion. The new title is “Pollen Grains Observed by SEM during Heating Process”.

(9) Line 60: “2.1.” instead of “2.1”. Apply to the entire document.

Response: We have revised this kind of error according to the reviewer’s suggestion.

(10) Lines 66, 68…..: Use Figure instead of Fig. according with the format of the journal. Apply to the entire document.

Response: We have replaced Fig. with Figure in the whole article according to the reviewer’s suggestion.

(11) Line 81: Capitalized Table.

Response: We have capitalized Table according to the reviewer’s suggestion.

(12) Figure 3: Increase the resolution of the figure.

Response: We have revised the Figure 3 according to the reviewer’s suggestion. We used the high resolution image instead of the old one.

(13) Line 144: Delete

Response: We have deleted the line 144 according to the reviewer’s suggestion.

(14) Figure 4: Axis: put separation before ( ).

Response: We have revised the Figure 4 according to the reviewer’s suggestion.

(15) Lines 153, 154, …….: Revise the format of ºC. Unify and apply to the entire document.

Response: We have revised the format of ºC and unified to the entire document.

(16) Table 2: Unify the format of tables. Delete most of the lines for separations. Volatiles instead of volatlles.

Response: We have unified the format of Table 2 and deleted the redundant lines. Meanwhile, we corrected the spelling mistake of Volatiles.

(17) Line 230: The title of the section is too long.

Response: We have changed the title of this section according to the reviewer’s suggestion. The new title is “Flavonoids Content Changes during Heating Process”.

(18) Figure 6: Unify the x axis: Time or t.

Response: We have revised the Figure 6 in the manuscript. In Figure 6A, the x axis represented the retention time (tR), while the x axis of Figure 6B was the heating time. So we added heating before time under the x axis of Figure 6B.

(19) Line 254: Capitalize Hui.

Response: We have capitalized Hui according to the reviewer’s suggestion.

(20) Lines 256, 261: Do not capitalize isorhamnetin and methanol.

Response: We have corrected those errors according to the reviewer’s suggestion.

(21) Lines 265, 266: Put a separation between the number and g.

Response: We have put a separation between the number and g.

(22) Line 268: The title of the section is too long.

Response: We have changed the title of this section according to the reviewer’s suggestion. The new title is “Pollen grains observed by SEM during heating process”.

(23) Sections 3.3, 3.4, …..: For each equipment add brand, city and country. Unify and apply to the entire document.

Response: We have added brand, city and country in sections 3.3, 3.4 and 3.7 according to the reviewer’s suggestion.

(24) Line 273: Delete the separation after “:”. Unify.

Response: We have deleted the separation after “:” according to the reviewer’s suggestion.

(25) Lines 287, 290, …….: Put separations after and before “=”. Apply to the entire document.

Response: We have put separations after and before “=” in the manuscript according to the reviewer’s suggestion.

(26) Section 3.7: Describe the compounds analyzed by HPLC. Indicate the range of each compound to prepare the calibration curve. Indicate r2, LOD, LOQ….

Response: We have added these results in the section 3.7 according to the reviewer’s suggestion. Table 7 showed the linear calibration curves, r2, and the LOD and LOQ of typhaneoside, isorhamnetin-3-O-neohespeidoside and isorhamnetin. The data obtained from the present HPLC method were considered to be satisfactory for subsequent analysis of all of the samples.

Table 7. Calibration curves, limits of detection (LOD), limits of quantitation (LOQ) of the three analytes.

Component

Calibration Curves

r2

Linear

(μg)

LOQ

(ng)

LOD

(ng)

Typhaneoside

0.9997

0.007516-0.4810

0.6413

0.4276

Isorhamnetin-3-O-neohespeidoside

0.9997

0.004838-0.3096

1.032

0.4128

Isorhamnetin

0.9997

0.002021-0.03234

1.0870

0.3623

(27) Line 343: ¿American?. USA.

Response: Sorry for this mistake. We have corrected this error and written “USA” in the manuscript.

(28) Line 346: water with 0.05% phosphoric acid.

Response: We have added “water with” before 0.05% phosphoric acid according to the reviewer’s suggestion.

(29) Line 345: Indicate the brand, city and country of the column.

Response: We have indicated the brand, city and country in the manuscript. The column is a chromatography PRAZIS Absolute C18 column (250 mm×4.6 mm, 5 μm) (Shanghai Dahu Scientific Instrument Co., Ltd., Shanghai, China).

(30) References: Use the format of the journal.

Response: Sorry for this mistake. We have revised the format of references.

 Once again, thank you very much for your comments and suggestions.

Reviewer 2 Report

This manuscript characterized the morphology and compounds changes, as well as determined the thermal properties changes during the carbonization process of pollen Typha. The authors analyzed the pyrolysis and combustion parameters of heated pollen Typha, indicating that thermal analysis techniques can be applied to the research on traditional Chinese medicine. This study is well organized and the data is reliable, some minor concerns are list as follows:

1. Introduction: what is the current research status regarding the application of thermal analysis in Traditional Chinese Medicine? 

2. Table 2, please explain the parameters such as L, a and b in the legend under the table.

3. Fig. 2B, please mark the time unit “min”.

4. Table 2, please explain the meaning of these parameters in the table legend under the table.

5. The English should be improved since there are many spelling mistakes, such as the “Typah” in line 14, page 1; line 138 “ether” in page 6.

Author Response

Dear Editor:

Thank you very much for your attention and the referee’s evaluation and comments on our manuscript entitled “Chemical Property Changes and Thermal Analysis during the Carbonizing Process of the Pollen Grains of Typha” (Manuscript ID: molecules-410664). We would like to express our sincere gratitude to the reviewer’s kindness and patience on reviewing our manuscript. We have now carefully revised our manuscript according to the reviewers’ comments and suggestions. A revised manuscript with the correction sections red marked was attached as the supplemental material and for easy check purpose. Should you have any questions, please contact us without hesitate.

Best regards,

Mingliang Gao

Name: Peidong Chen & Li Zhang

E-mail address: cpd@njucm.edu.cn (P.C.); zhangli@njucm.edu.cn (L.Z.)

2018-12-23

(1) Introduction: what is the current research status regarding the application of thermal analysis in Traditional Chinese Medicine?

Response: Thermal analysis has been used to monitor the physical or chemical properties changes during the process of heating [1,2]. It also can be used as a method for evaluating the process technology of Chinese medicine [3].

Reference:

1. Alain, F.P.; José M.; Fernández, J.L. Application of thermal analysis techniques in soil science. Geoderma 2009, 153, 1-10, doi:10.1016/j.geoderma.2009.08016.

2. Ma, J.; Meng, X.; Guo, X.; Lan, Y.; Zhang, S. Thermal analysis during partial carbonizing process of rhubarb, moutan and burnet. PLoS One 2017, 21, e0173946, doi:10.1371/journal.pone.0173946.

3. Meng, X.; He, M.; Guo, R.; Duan, R.; Huo, F.; Lv, C.; Wang, B.; Zhang, S. Investigation of the Effect of the Degree of Processing of Radix Rehmanniae Preparata (Shu Dihuang) on Shu Dihuangtan Carbonization Preparation Technology. Molecules 2017, 22, 1193, doi:10.3390/molecules22071193.

(2) Table 2, please explain the parameters such as L, a and b in the legend under the table.

Response: Thanks for your suggestion. We have added the meaning of L*, a* and b* under the Table 2. L*:the brightness of the color; a*:the degree of red or green; b*:the degree of yellow or blue.

(3) Fig. 2B, please mark the time unit “min”.

Response: Thanks for your suggestion. We have added “Heating Time” under the Figure 2B and marked the time unit “min”.

(4) Table 2, please explain the meaning of these parameters in the table legend under the table.

Response: Thanks for your suggestion. We have explained the meaning of these parameters in the manuscript. L* indicated brightness from black to white, a* and b* axis indicated the changing color. The positive direction of the a* axis represented the color to red, and the negative direction represented the color to green; the positive direction of b* axis represented the color to yellow and the negative direction represented to blue. ΔL* = value L* of specimens in the conventional layering system (CLS) group - value L* of specimens in digital veneering system (DVS) group; Δa* indicated the degree of red-green samples and Δa* = value a* of CLS - value a* of DVS; Δb* was the degree of yellow-blue samples and Δb* = value b* of the CLS - value b* in the DVS [1].

Reference: [1] Anghel, M.; Vlase, G.; Bilanin, M.; Vlase, T.; Albu, P.; Fuliaş, A.; Tolan, I.; Doca, N. Comparative study on the thermal behavior of two similar triterpenes from birch. J Therm Anal Calorim 2013, 113, 1379–1385, doi:10.1007/s10973-013-3203-3.

 (5) The English should be improved since there are many spelling mistakes, such as the “Typah” in line 14, page 1; line 138 “ether” in page 6.

Response: Sorry this mistake in line 14, page 1. We have revised this error in the manuscript. While the “ether” in line 138, page 6 isn’t a spelling mistake. The ether bond is a name of chemical bond. Besides we have improved the English in our manuscript and corrected many mistakes.

Once again, thank you very much for your comments and suggestions.

Round 2

Reviewer 1 Report

Dear editor,

This paper describes the chemical property changes and thermal analysis during the carbonizing process of the pollen grains of typha. The authors have made most of the indicated changes so the quality of the article has improved considerably. For this reason I believe that this work could be published in Molecules.